# A mitotic CDK5-PP4 phospho-signaling cascade primes 53BP1 for DNA repair in G1

Xiao-Feng Zheng[1], Sanket S. Acharya[1], Katherine N. Choe[1], Kumar Nikhil[2], Guillaume Adelmant [3,4,5], Shakti Ranjan Satapathy[2], Samanta Sharma[3,6], Keith Viccaro[2], Sandeep Rana[7], Amarnath Natarajan [7,8], Peter Sicinski [3,6], Jarrod A. Marto [3,4,5,9], Kavita Shah [2] & Dipanjan Chowdhury [1,10,11]

Mitotic cells attenuate the DNA damage response (DDR) by phosphorylating 53BP1, a critical DDR mediator, to prevent its localization to damaged chromatin. Timely dephosphorylation of 53BP1 is critical for genome integrity, as premature recruitment of 53BP1 to DNA lesions impairs mitotic fidelity. Protein phosphatase 4 (PP4) dephosphorylates 53BP1 in late mitosis to allow its recruitment to DNA lesions in G1. How cells appropriately dephosphorylate 53BP1, thereby restoring DDR, is unclear. Here, we elucidate the underlying mechanism of kinetic control of 53BP1 dephosphorylation in mitosis. We demonstrate that CDK5, a kinase primarily functional in post-mitotic neurons, is active in late mitotic phases in non-neuronal cells and directly phosphorylates PP4R3β, the PP4 regulatory subunit that recognizes 53BP1. Specific inhibition of CDK5 in mitosis abrogates PP4R3β phosphorylation and abolishes its recognition and dephosphorylation of 53BP1, ultimately preventing the localization of 53BP1 to damaged chromatin. Our results establish CDK5 as a regulator of 53BP1 recruitment.

[1] Division of Radiation and Genome Stability, Department of Radiation Oncology, Dana-Farber Cancer Institute, Harvard Medical School, Boston, MA 02215, USA. [2] Department of Chemistry and Purdue University Center for Cancer Research, Purdue University, West Lafayette, IN 47907, USA. [3] Department of Cancer Biology, Dana-Farber Cancer Institute, Boston, MA 02215, USA. [4] Blais Proteomics Center, Dana-Farber Cancer Institute, Boston, MA 02115, USA. [5] Department of Pathology, Brigham and Women's Hospital and Harvard Medical School, Boston, MA 02115, USA. [6] Department of Genetics, Blavatnik Institute, Harvard Medical School, Boston, MA 02115, USA. [7] Eppley Institute for Research in Cancer, University of Nebraska Medical Center, Omaha, NE 68198, USA. [8] Fred and Pamela Buffett Cancer Center, University of Nebraska Medical Center, Omaha, NE 68198, USA. [9] Department of Oncologic Pathology, Dana-Farber Cancer Institute, Boston, MA 02115, USA. [10] Broad Institute of Harvard and MIT, Cambridge, MA 02142, USA. [11] Department of Biological Chemistry and Molecular Pharmacology, Harvard Medical School, Boston, MA 02115, USA. Correspondence and requests for materials should be addressed to D.C. (email: dipanjan_chowdhury@dfci.harvard.edu)

53BP1 is a mediator of DNA double-strand break (DSB) repair whose recruitment to chromatin is regulated throughout cell cycle[1]. During mitosis, phosphorylation of threonine 1609 (T1609) and serine 1618 (S1618) in the ubiquitination-dependent recruitment (UDR) motif of 53BP1 blocks its interaction with damage-induced ubiquitylated-H2AK15 (H2AK15ub), impeding 53BP1 from binding to damaged chromatin[2,3]. Protein phosphatase 4 (PP4), in complex with its regulatory subunit PP4R3β, dephosphorylates T1609/S1618 of 53BP1 to allow 53BP1 binding to damaged chromatin during G1[2]. Timely dephosphorylation of 53BP1 UDR motif is critical for genome integrity, as premature recruitment of phospho-null 53BP1 to mitotic DNA lesions is associated with mitotic defects, aneuploidy, and telomere fusions[2,3]. Conversely, constitutive phosphorylation of 53BP1 prevents its localization to DNA damage site in G1[2].

PP4 belongs to the PP2A family of serine/threonine protein phosphatases whose members regulate mitotic exit and DNA repair[4,5]. PP4 is constitutively expressed in the cytoplasm and nucleus, and is active in all stages of the cell cycle[6]. Substrate specificity for PP4 is determined by its interaction with regulatory subunits PP4R1, PP4R2, PP4R3α, PP4R3β, and PP4R4, each of which recognizes distinct substrates under various physiological contexts[7–10]. PP4R3β and 53BP1 interact in mitosis; and the depletion of PP4R3β impairs the dephosphorylation of 53BP1[2].

We investigated the regulation of PP4R3β-53BP1 interaction during mitosis that ensures proper timing of 53BP1 dephosphorylation, and found a phosphorylation site, serine 840 (S840) in the 53BP1-interacting region of PP4R3β. We show that phosphorylation of S840 is required for PP4R3β interaction with 53BP1. Intriguingly, CDK5, an atypical non-cyclin-dependent kinase, phosphorylates PP4R3β at S840.

CDK5 is a serine/threonine kinase that is structurally similar to CDK2[11], but unlike canonical CDKs, it is not activated by cyclins. The co-factors for CDK5 activity are p35 or p39, and their respective cleaved counterparts p25 and p29, which are abundant in post-mitotic neurons[12]. Expectedly, CDK5 is highly expressed in the brain and its function is critical for neuronal survival and brain development[13]. CDK5 has also been implicated in DNA damage-induced neuronal cell death[14,15]. In contrast, limited studies with cancer cells have shown that CDK5 promotes survival and proliferation[16]. Presently, there is still a very limited understanding of CDK5 function in non-neuronal cells.

Here, we observed that CDK5 is active during mitosis of non-neuronal cells. The kinetics of CDK5 activity correspond with PP4R3β phosphorylation at S840 and with 53BP1 dephosphorylation. Specific inhibition of CDK5 recapitulates PP4R3β S840 phosphonull phenotypes, in terms of deficient interaction with 53BP1, dephosphorylation of 53BP1 at T1609 and S1618, irradiation-induced foci formation (IRIF), and aphidicolin-induced nuclear body formation of 53BP1. Conversely, expression of the S840D phosphomimetic variant of PP4R3β rescues 53BP1 functional defects that arise from CDK5 inhibition. Therefore, we report that CDK5 acts in the DDR of non-neuronal cells by regulating the localization of 53BP1 to sites of DNA damage.

## Results

### Phosphorylation of PP4R3β promotes interaction with 53BP1.
Among the aforementioned five known PP4 regulatory subunits, only the depletion of PP4R3β renders a defect in 53BP1 dephosphorylation[2]. We therefore characterized the interaction of PP4R3β with 53BP1. We immunoprecipitated 53BP1 with full-length or deletion mutants of PP4R3β, and identified that the C-terminal region (aa 721–849) is both necessary and sufficient to interact with 53BP1 (Supplementary Fig. 1a). Since phosphorylation dynamics regulate mitotic progression, we considered that phosphorylation of PP4R3β C-terminal region may regulate its interaction with 53BP1. Mass spectrometry (MS) analysis of phospho-peptides of PP4R3β purified from mitotic cells (Supplementary Fig. 1b) revealed serine 840 (S840) to be the only residue phosphorylated (Fig. 1a). Phosphorylation state and site assignment for S840 were verified using a synthetic phospho-peptide (Supplementary Fig. 1c). S840 is conserved among vertebrate species (Supplementary Fig. 1d) as a proline-directed phosphorylation site in the C-terminal region unique to PP4R3β (Supplementary Fig. 1e). We next mined cBioPortal Cancer Genomics database[17] for cancer-relevant variants of S840, and identified a serine-to-phenylalanine melanoma variant, S840F[18]. Therefore, this residue is likely to be physiologically relevant.

We evaluated how phosphorylation of S840 influences the PP4R3β interaction with 53BP1 in mitotic cells relative to asynchronous cells. The PP4R3β and 53BP1 association was weak in asynchronous cells relative to cells arrested in mitosis. 53BP1 co-immunoprecipitated with wild-type PP4R3β and phosphomimetic variant S840D at comparable levels in mitotic cells; however, its interaction was reduced with phosphonull variants S840F and S840A (Fig. 1b, FLAG IP panel). Endogenous PP4 catalytic subunit (PP4C) co-immunoprecipitated efficiently with all S840 phospho-variants of PP4R3β under all conditions (Fig. 1b, MYC IP panel). These results suggest that phosphorylation of PP4R3β on S840 is required for its interaction specifically with 53BP1 but is dispensable for the integrity of the phosphatase holoenzyme complex.

### 53BP1 localization to DSBs requires PP4R3β phosphorylation.
We next examined whether phosphorylation of PP4R3β at S840 is required for the dephosphorylation of 53BP1 at T1609/S1618. Using a phospho-specific antibody against T1609/S1618[2], we monitored the phosphorylation status of these residues on 53BP1, in cells where endogenous PP4R3β was replaced with Myc-tagged wild-type or S840F variant (Supplementary Fig. 1i). During the transition from mitosis to G1(Supplementary Fig. 1j), 53BP1 phosphorylation at T1609/S1618 dissipated in control siRNA-treated cells and in cells expressing wild-type PP4R3β but persisted in cells depleted of endogenous PP4R3β and in cells expressing S840F (Fig. 1c). These results indicate that phosphorylation of PP4R3β is a prerequisite for the dephosphorylation of 53BP1 at T1609 and S1618. Since PP4C/PP4R3β-mediated dephosphorylation of 53BP1 is necessary for its localization to DSB repair foci in G1 cells[2], we reasoned that absence of PP4R3β S840 phosphorylation will also preclude irradiation-induced 53BP1 foci formation. Indeed, irradiated G1 cells expressing S840F and cells depleted of PP4R3β exhibited a 2-fold reduction of 53BP1 foci compared to cells expressing the wild-type PP4R3β (Fig. 1d, e). These results suggest that dephosphorylation of 53BP1 during mitosis depends on the phosphorylation of PP4R3β at S840 to allow 53BP1 recruitment to damaged chromatin.

### PP4R3β phosphorylation impacts cancer therapy.
53BP1-deficient cells are radiosensitive and loss of 53BP1 renders BRCA1-null cells resistant to PARP inhibitors[19–21]. Therefore, we asked whether phosphorylation of PP4R3β may regulate cellular sensitivity to radiation and response to PARP inhibition in BRCA1-null cells. Cells expressing PP4R3β S840F are ~1.5 fold more radiosensitive than those expressing wild-type (Fig. 1f). BRCA1-null ovarian UWB1.289 cells expressing PP4R3β S840F are ~2 fold more resistant toward clinical grade PARP inhibitor Olaparib than those expressing wild-type PP4R3β (Fig. 1g). Together, these results highlight that the phosphorylation of

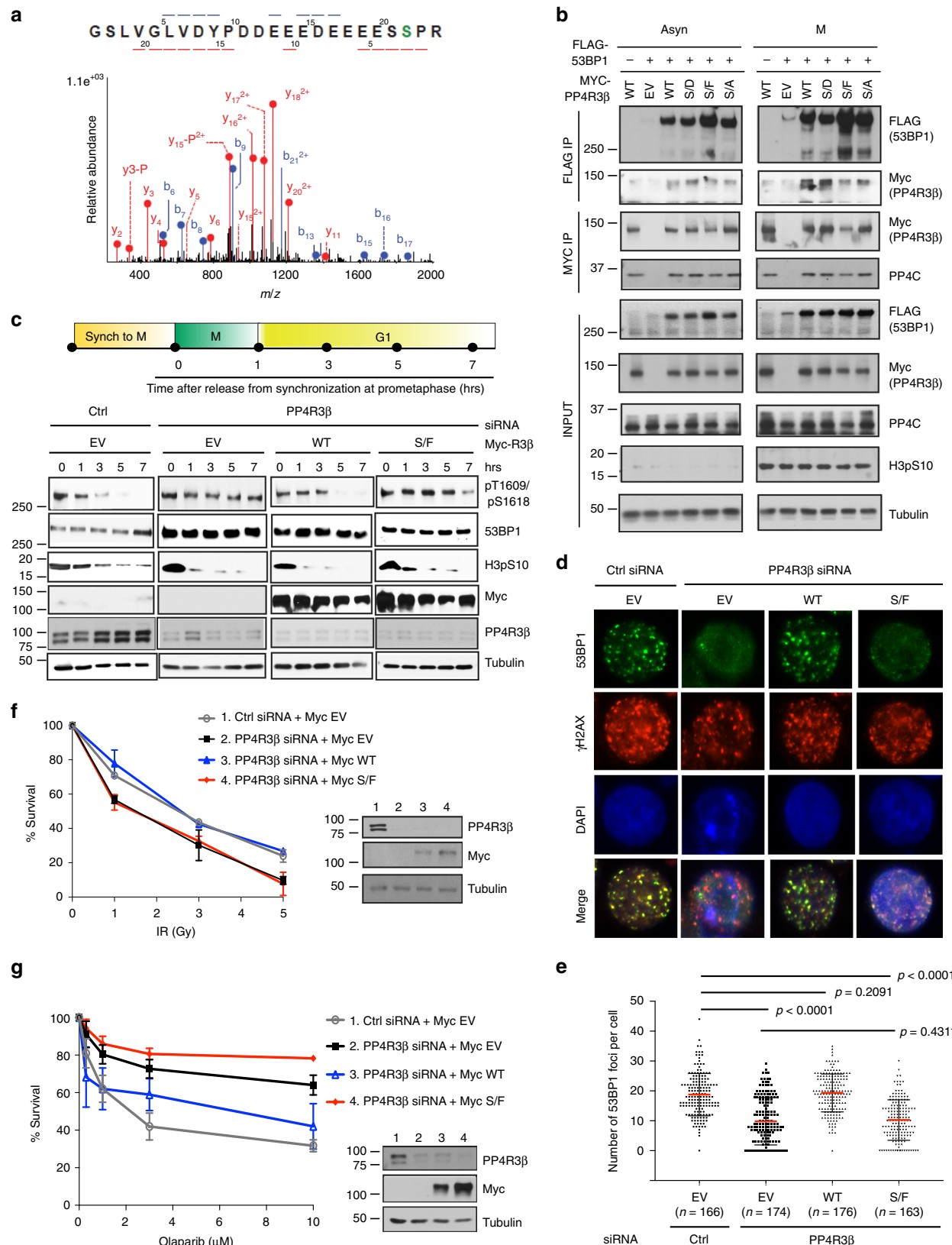

PP4R3β at S840 impacts the DNA repair mediator function of 53BP1. Furthermore, these results suggest that this phospho-signaling axis is relevant for cancer therapy.

**Phosphorylation of 53BP1 and PP4R3β during mitosis.** Since 53BP1 dephosphorylation and function are dependent on PP4R3β S840 phosphorylation, we investigated the temporal correlation between 53BP1 phosphorylation at T1609/S1618 and PP4R3β phosphorylation at S840 throughout mitosis. We synchronized cells to the G2/M border prior to mitotic entry using RO-3306[22], and monitored the phosphorylation of these sites at distinct stages of mitosis after cells were released from

**Fig. 1** PP4R3β S840 phosphorylation is required for 53BP1 recruitment to damaged chromatin. **a** MS/MS spectrum of the PP4R3β phospho-peptide harboring S840. Phosphorylated residue is highlighted in green. **b** Co-immunoprecipitation of 53BP1 or PP4C with PP4R3β S840 phospho-variants. HeLa cells transiently transfected with the indicated FLAG-53BP1 and Myc-PP4R3β constructs were left in an asynchronous state (Asyn) or synchronized to prometaphase (M) by treatment with nocodazole (Supplementary Fig. 1f–h). Whole cell extracts were immunoprecipitated (IP) with indicated agarose beads and analyzed by immunoblot using indicated antibodies. EV: empty vector; S/D: S840D; S/F: S840F; S/A: S840A. **c** Immunoblots showing the kinetics of 53BP1 phosphorylation throughout mitosis in cells expressing PP4R3β S840 phospho-variants . HeLa cells were transfected with the indicated siRNAs and complemented with the indicated siRNA-resistant Myc-PP4R3β S840 phospho-variant constructs. Transfected cells were released from prometaphase arrest and harvested at indicated time points for immunoblot analysis using indicated antibodies. Phospho-histone H3 at Ser10 (H3pS10) indicates mitotic state. **d** Irradiation-induced 53BP1 foci formation in cells expressing PP4R3β S840 phospho-variants. Mitotic HeLa cells were complemented with Myc-PP4R3β S840 phospho-variant constructs as described in **c**, collected from prometaphase arrest and seeded on poly-D-lysine-coated coverslips. Four hours after release, cells were irradiated with 5 Gy. Two hours post-irradiation, cells were fixed and analyzed by immunofluorescence with indicated antibodies. γ-H2AX staining marks sites of DNA damage. **e** Quantifications of 53BP1 foci in **d**, pooled from triplicate repeat experiments, expressed as mean ± s.d. Total number of cells is indicated in parenthesis. P-values, Mann–Whitney U Test. **f** Radiosensitivity of cells expressing PP4R3β S840 phospho-variants. HeLa cells were complemented with siRNA-resistant Myc-PP4R3β S840 phospho-variant constructs as described in **c**. Viability was evaluated by clonogenic survival. Immunoblots confirm siRNA efficiency and expression of siRNA-resistant constructs. Data are expressed as mean ± s.d; n = 3. **g** Response of BRCA1-null UWB1.289 cells expressing PP4R3β S840 phospho-variants to indicated concentration of clinical-grade PARP inhibitor Olaparib. Cells were complemented as described in **c**. Viability was assessed by CellTiter-Glo colorimetric assay. Immunoblots confirm siRNA efficiency and expression of siRNA-resistant constructs. Data are expressed as mean ± s.d; n = 3

RO-3306-induced arrest. We confirmed that phospho-specific antibodies against T1609/S1618 of 53BP1[2] and S840 of PP4R3β (Supplementary Fig. 2a–d) detected an enrichment of phosphorylated 53BP1 (Supplementary Fig. 2a) and phosphorylated PP4R3β (Supplementary Fig. 2e, f) in mitotic cells. 53BP1 T1609/S1618 phosphorylation is relatively low in early mitosis (prophase), then sharply rises in prometaphase and metaphase, before dropping to the same basal level as that of prophase, during the late stages of mitosis (Fig. 2a, b). The kinetics of 53BP1 phosphorylation is consistent with the activity of CDK1, PLK1, and p38 MAPK[23], which were identified as kinases that phosphorylate T1609 and S1618 residues of the 53BP1[2,3]. In contrast, PP4R3β S840 phosphorylation increased upon release from G2/M and peaked during metaphase and anaphase, before declining in telophase to return to near basal levels in G1 (Fig. 2c, d). Relative to the diminished 53BP1 T1609/S1618 phosphorylation, the persistent PP4R3β S840 phosphorylation during anaphase and telophase suggests that the kinase(s) responsible for phosphorylation of PP4R3β may be a key regulator of the subsequent PP4C/PP4R3β-mediated dephosphorylation of 53BP1.

**CDK5 is active in mitosis.** Scansite[24] algorithm predicted CDK1 and CDK5 as proline-directed serine/threonine kinases to phosphorylate S840, with CDK5 being the more likely of the two kinases (Supplementary Fig. 3a). CDK5 is primarily activated by cyclin-like proteins p35, p25, p39, and p29, which are abundant in post-mitotic neurons[12,25]. Considering that phosphorylation of PP4R3β specifically occurs in mitosis, we examined whether CDK5 is active in mitosis and targets PP4R3β at S840. We quantified the kinase activity of immunoprecipitated CDK5 from HeLa cells undergoing mitosis upon release from RO-3306-induced arrest at G2/M (Fig. 2e, f; Supplementary Fig. 3b, c). In parallel we monitored PP4R3β phosphorylation at S840 and 53BP1 phosphorylation at T1609/ S1618 at the same time points (Fig. 2g). Consistent with PP4R3β phosphorylation kinetics, CDK5 displayed increasing kinase activity as cells progressed through mitosis and remained highly active in the late stages of mitosis (Fig. 2f). CDK1 kinase activity mirrored the 53BP1 phosphorylation at T1609/ S1618, with a sharp decline to basal levels in late mitosis, while CDK5 activity level remained high (Supplementary Fig. 3b). These results implicate CDK5 as a kinase that phosphorylates PP4R3β at S840.

**PP4R3β is a bona fide substrate of CDK5.** To assess whether PP4R3β is a substrate of CDK5, we tested whether CDK5

phosphorylates PP4R3β in vitro. Indeed, purified CDK5/p25 complex phosphorylated recombinant His-PP4R3β C-terminal fragment (aa 721–849) but not the PP4R3β S840A phosphonull counterpart (Fig. 3a, lane 3), indicating that CDK5 directly targets S840 of PP4R3β. Next, we examined the association of PP4R3β with CDK5 in cells and observed that FLAG-HA-PP4R3β interacts with endogenous CDK5 during mitosis (Fig. 3b). These results suggest that CDK5 binds and phosphorylates PP4R3β during mitosis.

To determine if CDK5-mediated phosphorylation of PP4R3β S840 impacts 53BP1 capacity in DNA repair, we quantified irradiation-induced 53BP1 foci formation in cells depleted of CDK5 (Supplementary Fig. 3d-g) and in cells treated with 20–223, an ATP-competitive catalytic inhibitor of CDK5[26] (Fig. 3c, d; Supplementary Fig. 3h, i). In both cases, there were at least a 2-fold decrease in 53BP1 foci per cell in CDK5-depleted or CDK5-inhibited cells relative to untreated cells. Moreover, consistent with cells expressing the PP4R3β S840F phosphonull variant, cells treated with 20–223 manifested a 1.5-fold increase in radiosensitivity and a 1.5-fold increase in resistance to PARP inhibitor Olaparib (Fig. 3e, f). However, as siRNA depletion takes effect only after several rounds of cell division, and an ATP-competitive catalytic inhibitor lacks specificity, we cannot formally rule out that the phenotype observed from these treatments may be due to inhibition of kinases other than CDK5 that are active in dividing cells.

To definitively establish that specific inhibition of CDK5 impacts phosphorylation of PP4R3β S840 precisely during mitosis, we employed a chemical genetic system that expresses an analog-sensitive (AS) variant of CDK5. The AS variant confers a structural change proximal to the catalytic site of the kinase, thus exhibiting wild-type kinase activity until it is inhibited upon binding to a bulky ATP analog that accommodates the mutated structure (Supplementary Fig. 4a)[27]. This system allows specific inhibition of CDK5 during mitosis. To ensure consistency between the observed phenotypes due to specific CDK5 inhibition and those observed from cells expressing the PP4R3β S840F variant identified in a melanoma patient, we replaced the endogenous CDK5 with a CDK5-AS variant in the malignant melanoma cell line A375 (Supplementary Fig. 4b). We confirmed that CDK5 in these AS cells has the same kinase activity kinetics as that of endogenous CDK5 (Supplementary Fig. 4c-g; Fig. 2f). Likewise, the endogenous CDK1 activity kinetics in these cells remains unaltered (Supplementary Fig. 4f).

To determine whether CDK5 phosphorylates PP4R3β at S840 specifically during mitosis, mitotic A375-AS cells collected

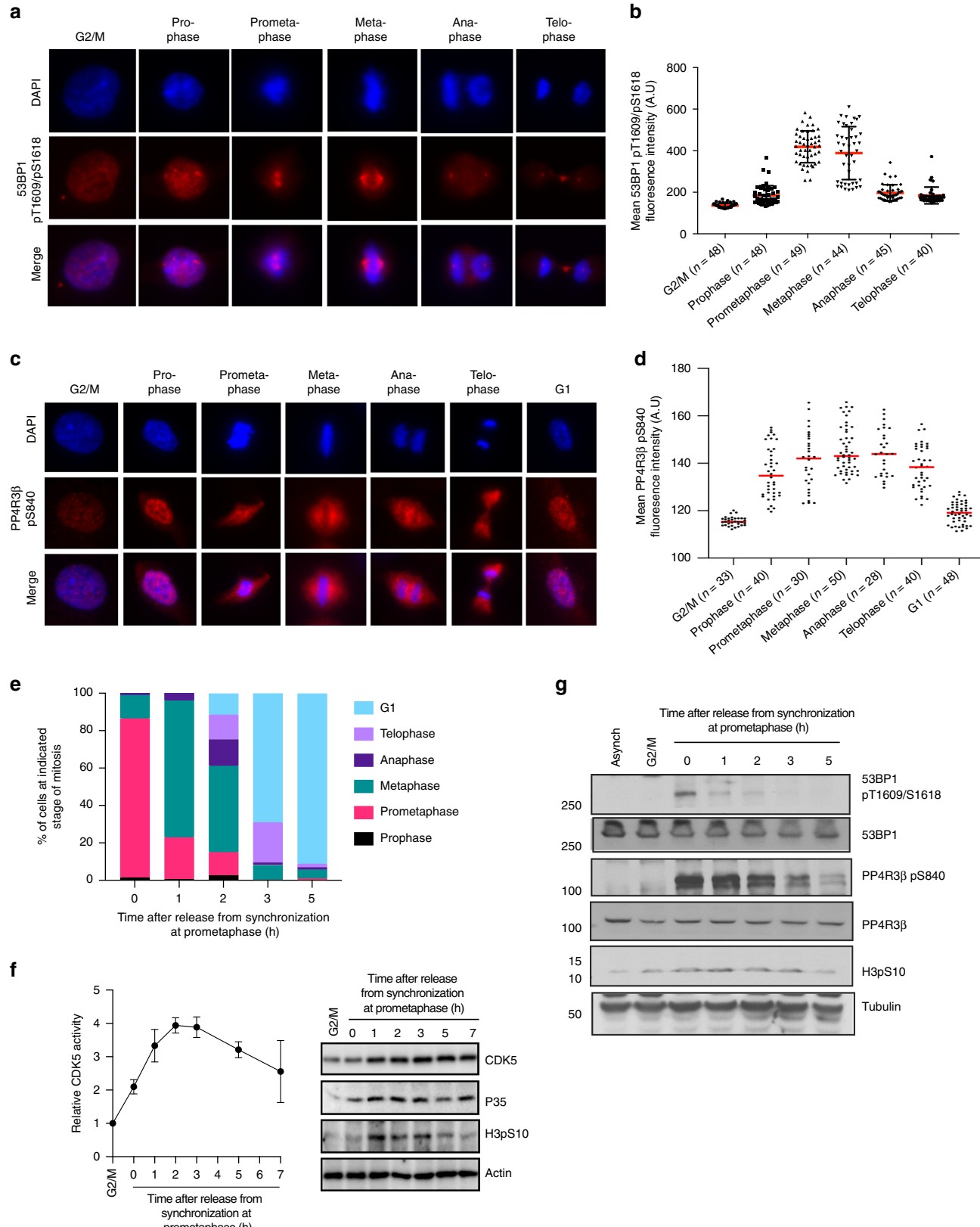

from nocodazole-induced arrest at prometaphase were released into fresh medium containing DMSO or bulky ATP-analog inhibitor 1NMPP1 (Supplementary Fig. 4h). Sample aliquots were then collected at various time points to monitor PP4R3β S840 phosphorylation during mitosis. CDK5 inhibition by treatment with 1NMPP1 abrogated PP4R3β S840 phosphorylation from 30 min onward after release from nocodazole-induced arrest (Fig. 3g, h). Collectively, these results indicate that PP4R3β is indeed a bona fide substrate of CDK5 during mitosis.

**CDK5 regulates 53BP1 recruitment to damaged chromatin.** Since specific inhibition of CDK5 abrogates phosphorylation of

**Fig. 2** CDK5 is active in mitosis. **a** Phosphorylation of 53BP1 T1609/S1618 at distinct stages of mitosis and in G1. RPE1 cells were fixed at 0, 15, 30, 60, 90, and 120 min after release from G2/M arrest and stained with phospho-specific antibody against T1609/S1618 (pT1609/pS1618). Cells at indicated stages of mitosis were selected based on chromatin morphology, as indicated by DAPI staining. **b** Fluorescence intensity of phosphorylated 53BP1 over area of interest was quantified. Staining intensities from number of events per stage (indicated in the parenthesis) were expressed as mean ± s.d. A.U: arbitrary unit. **c** Phosphorylation of PP4R3β S840 at distinct stages of mitosis and in G1. RPE1 cells were synchronized and fixed at time points after release as described in **a** and stained with a phospho-specific antibody against S840 (pS840). **d** Fluorescence intensity of phosphorylated PP4R3β over area of interest was quantified as described in **b**. **e** Distribution of HeLa cells across distinct stages of mitosis at indicated time points after release from prometaphase arrest. HeLa cells were arrested at G2/M by RO-3306 and released into nocodazole to enrich for cells in prometaphase for 1 h. Prometaphase-arrested cells were collected by shake-off and released to progress through mitosis into G1 and fixed at indicated time points. Cells corresponding to stages of mitosis were tallied based on chromatin morphology, as indicated by DAPI staining. **f** Kinase activity levels of CDK5 in mitosis and G1. HeLa cells were synchronized and released as described in **e**. CDK5, immunoprecipitated from cells collected at indicated time points after release, was incubated with substrate peptide and [$^{32}$P]-ATP. Radioactivity of labeled substrate peptide was measured in a liquid scintillation counter. Data are expressed as mean ± s.d; $n = 3$. Representative immunoblots show indicated proteins present in the cell lysate for kinase assay. **g** Immunoblots showing the kinetics of 53BP1 T1609/S1618 phosphorylation and PP4R3β S840 phosphorylation throughout mitosis. HeLa cells were collected at indicated time points after release from synchronization as described in **e** and were analyzed with indicated antibodies

PP4R3β on S840, we reasoned that specific inhibition of CDK5 will also prevent PP4R3β recognition of 53BP1 and the dephosphorylation of 53BP1 on T1609 and S1618. Indeed, inhibition of CDK5 by treatment with 1NMPP1 also abolished 53BP1-PP4R3β interaction (Fig. 4a) and, consequently, impaired dephosphorylation of 53BP1 at T1609/S1618 (Fig. 4b). Next, we examined the recruitment of 53BP1 to DNA lesions and observed a ~2-fold reduction of irradiation-induced 53BP1 foci in G1 cells where CDK5 is inhibited by 1NMPP1 (Fig. 4c, d).

Distinct nuclear bodies visible in G1 cells represent endogenous DNA damage that occurs during DNA replication and are carried through mitosis[28,29]. 53BP1 is recruited to nuclear bodies in G1 cells and potentially helps to resolve these DNA lesions[29]. Low doses of aphidicolin increase the frequency of nuclear bodies in cells without triggering cellular checkpoints. Consistent with previous results, inhibition of CDK5 in A375-AS cells compromised the number of 53BP1-positive nuclear bodies (Fig. 4e, f). Together, these results suggest that a phospho-signaling cascade initiated by CDK5 and mediated by PP4R3β regulates the recruitment of 53BP1 to damaged chromatin.

**KIAA0528 is required for 53BP1 localization to DNA damage.** We observed that CDK5 was becoming increasingly active as cells progressed through mitosis (Fig. 2f). Moreover, we noted that CDK1 activity levels correlated with protein levels of CDK1 and its activator cyclin B during the course of mitosis, which peaked during metaphase and diminished thereafter (Supplementary Fig. 3b, c). However, levels of CDK5 and its established post-mitotic activator p35 stayed relatively constant during the course of mitosis (Fig. 2f; Supplementary Fig. 3c; Supplementary Fig. 4f, g). Therefore, it is very likely that alternative CDK5 activator(s) stimulates CDK5 kinase activity specifically during mitosis in non-neuronal cells. Fibroblast growth factor (acidic) intracellular binding protein (FIBP) and KIAA0528 (C2 calcium-dependent domain containing 5) have recently been identified as CDK5 binding partners in non-neuronal cells, and are required for growth and migration of breast cancer cells[30]. We therefore tested whether CDK5 could interact with p35, FIBP, or KIAA0528 under cellular conditions in which CDK5 is highly active. Pulldown of endogenous CDK5 from asynchronous cells and from cells in late stages of mitosis, when CDK5 activity is high (Supplementary Fig. 3c), did not co-immunoprecipitate p35 or FIBP, but did co-immunoprecipitate KIAA0528 (Supplementary Fig. 5a). Depletion of KIAA0528 also rendered persistent 53BP1 phosphorylation at T1609/S1618 (Supplementary Fig. 5b), which caused a defect in irradiation-induced 53BP1 foci formation (Supplementary Fig. 5c, d). Collectively, these results suggest that

KIAA0528 is a possible CDK5 co-factor that is required for the recruitment of 53BP1 to DNA damage in non-neuronal cells.

**CDK5-PP4 signaling axis drives 53BP1 to damaged chromatin.** To confirm that the impact of CDK5 on 53BP1 recruitment was due to the phosphorylation of S840 on PP4R3β, we tested whether the effect of CDK5 inhibition could be reversed by the expression of PP4R3β phosphomimetic variant S840D. We overexpressed PP4R3β S840 phospho-variants in A375-AS cells, then treated these cells with 1NMPP1 to examine the formation of 53BP1 foci upon irradiation in G1, or nuclear bodies upon treatment with aphidicolin. Only the expression of phosphomimetic variant S840D rescued the CDK5 inhibition-induced defect in 53BP1 irradiation foci formation (Fig. 5a, b) and nuclear body formation (Fig. 5c, d). We thus establish a cell cycle-regulated phospho-signaling cascade comprised of CDK5 and PP4/PP4R3β that drives the recruitment of 53BP1 to damaged chromatin.

### Discussion
Based on our data, we envision a spatiotemporal model (Fig. 5e) in which, during prometaphase and metaphase, CDK1 and PLK1 phosphorylate 53BP1 at T1609 and S1618 in the UDR motif, preventing 53BP1 from recognizing DNA damage-induced modified histones[3,31]. CDK5 phosphorylates PP4R3β at S840 starting at the same stages of mitosis, and both CDK5 activity and PP4R3β S840 phosphorylation increase and persist throughout mitosis. During anaphase and telophase, phosphorylated active PP4R3β recognizes 53BP1 and catalyzes the dephosphorylation of 53BP1 through PP4C. Upon exposure to DSBs, dephosphorylated 53BP1 readily recognizes DNA damage-induced ubiquitylated H2AK15 in G1. It is noteworthy that recruitment of 53BP1 to DSBs is a complex process and the CDK5/PP4 axis works in conjunction with other factors to modulate this important step in the DDR.

53BP1 is pivotal for DSB repair pathway choice, with ramifications in DNA damaging cancer therapy and overall genomic stability. Contextual phosphorylation of 53BP1 regulates its function in DSB repair. ATM-mediated phosphorylation of Ser/Thr-Gln (S/T-Q) motifs in the N-terminus is necessary for 53BP1 activity in interphase[32], whereas CDK1/PLK1-mediated phosphorylation of T1609/S1618 residues of 53BP1 impedes its function in mitosis[3]. Sequestering 53BP1 from damaged chromatin during mitosis is necessary for genomic stability. A key mechanism in restoring 53BP1 recruitment to damaged chromatin in G1 is the PP4C/PP4R3β-mediated dephosphorylation of 53BP1during late mitosis. We report here that the interaction between PP4R3β and 53BP1,

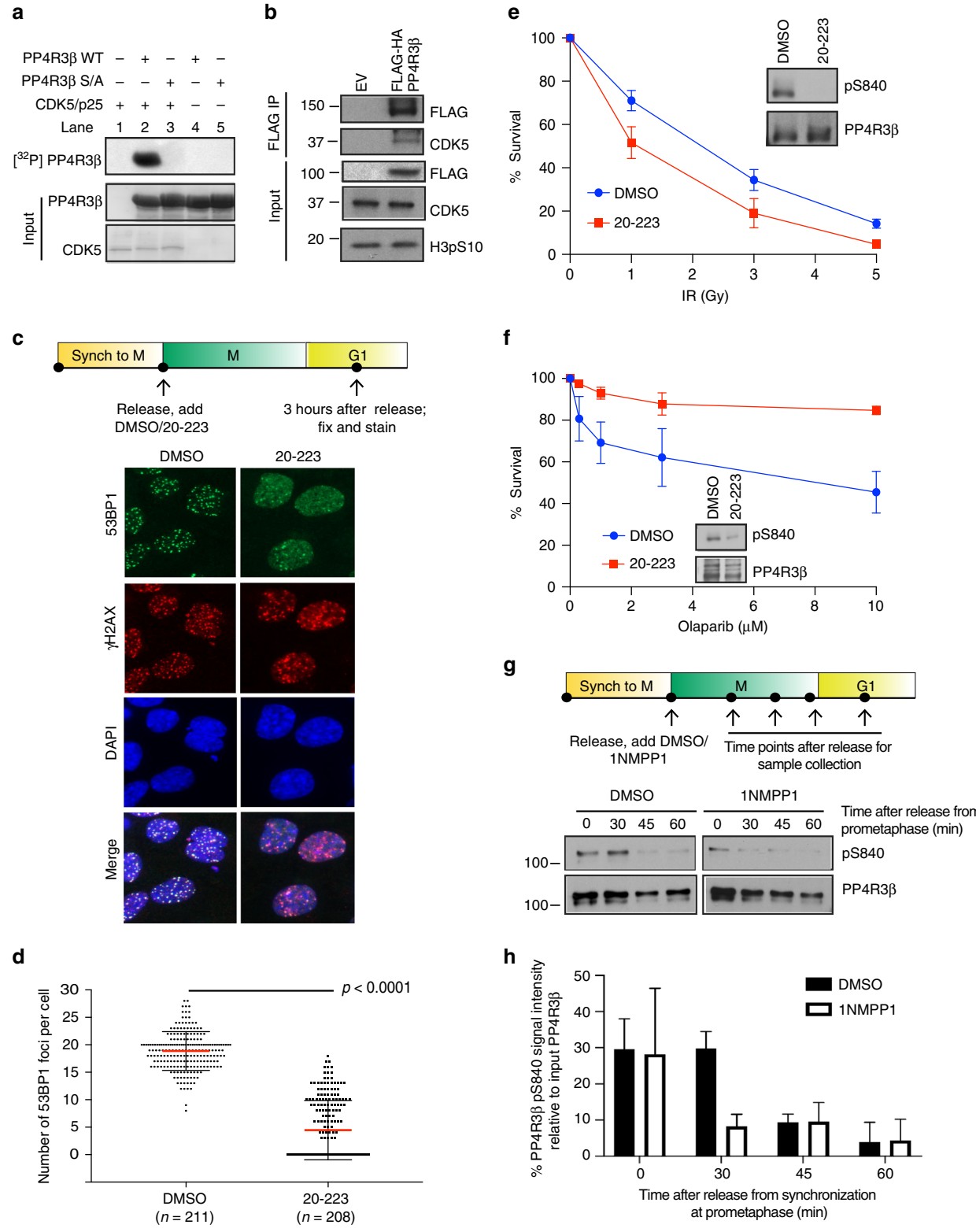

and the subsequent dephosphorylation of 53BP1, are dependent on the phosphorylation of S840 of PP4R3β.

Our results demonstrate that a single post-translational modification on a PP2A-family of serine/threonine protein phosphatase regulatory subunit impacts the substrate specificity of the phosphatase complex. Post-translational modifications on the catalytic subunits of the PP2A family are well studied[33] and the impact is pleiotropic. For example, leucine carboxyl

methyltransferase (LCMT-1)-mediated carboxyl methylation of PP4C is required for the formation of PP4-PP4R1 holoenzyme; and methylation of PP2A is required for the formation of PP2A holoenzyme containing Bα and Bβ-type regulatory subunits[34,35]. Loss of LCMT-1 reduces PP2A holoenzyme formation and causes severe defects in fetal hematopoiesis and, consequently, embryonic lethality in mice[36]. We speculate that the modification of a phosphatase regulatory subunit is more likely to regulate a

**Fig. 3** PP4R3β is a bona fide substrate of CDK5. **a** Reconstituted CDK5/p25 complex was incubated with purified wild-type or S840A PP4R3β. $^{32}$P-ATP signal indicates CDK5/p25-mediated phosphorylation. Ponceau-stained blots show input proteins. **b** HeLa cells stably expressing FLAG-HA empty vector (EV) or FLAG-HA PP4R3β were arrested in prometaphase, lysed, and immunoprecipitated for FLAG-HA PP4R3β. Elution from FLAG immunoprecipitation was analyzed by immunoblot with indicated antibodies. **c** RPE1 cells released from RO-3306-induced G2/M arrest were immediately treated with DMSO or 10 μM of CDK5 inhibitor compound 20-223. One hour after release from RO-3306, prometaphase cells were collected by shake-off and seeded on poly-D-lysine-coated coverslips. Three hours after release, cells were irradiated with 5 Gy IR. Two hours post-irradiation, cells were fixed and stained with indicated antibodies to detect irradiation-induced 53BP1 foci. γ-H2AX staining marks sites of DNA damage. **d** Quantifications of 53BP1 foci in **c**, pooled from triplicate repeat experiments, expressed as mean ± s.d. Total number of cells is indicated in parenthesis. P-values, Mann–Whitney U Test. **e** Effect of CDK5 inhibition on radiosensitivity. Hela cells were treated with DMSO or 250 nM of 20-223 for 48 h, followed by irradiation at the indicated doses. Cell viability was evaluated by clonogenic survival. Data are expressed as mean ± s.d; n = 3. Immunoblots indicate abrogation of PP4R3β S840 phosphorylation from treatment with 20-223. **f** Response of BRCA1-null UWB1.289 cells treated with DMSO or 250 nM of 20-223 to indicated concentration of clinical-grade PARP inhibitor Olaparib. Cell viability was assessed by CellTiter-Glo assay. Data are expressed as mean ± s.d; n = 3. Immunoblots indicate abrogation of PP4R3β S840 phosphorylation from treatment with 20-223. **g** Melanoma A375 cells stably expressing CDK5 analog-sensitive variant (A375-AS) were arrested in prometaphase. Mitotic cells were released into media containing either DMSO or 10 μM of non-hydrolyzable ATP analog, 1NMPP1, to specifically inhibit CDK5. Cells were harvested at indicated time points for immunoblot analysis with indicated antibodies. **h** Quantification of immunoblot intensity in **g**, from duplicate repeat experiments, by ImageJ

more defined subset of substrates, thereby impacting specific signaling pathway(s). Our findings herein unveil an additional regulatory layer to phospho-signaling cascades. Future studies on phosphorylation of S840 of the PP4R3β would reveal whether the impact of this post-translational modification extends beyond 53BP1 and DDR signaling.

We identified CDK5 as a kinase that phosphorylates PP4R3β. Biochemical quantification of CDK5 kinase activity in different stages of mitosis provides direct evidence that CDK5 is indeed active during mitosis in non-neuronal cells and is an important player in the phospho-signaling cascade for 53BP1 recruitment to sites of DNA damage. While our data strongly suggests CDK5 phosphorylates PP4R3β, it is also quite possible that other mitotic kinases, such as CDK1, might also phosphorylate PP4R3β, especially during the early-to-mid stages of mitosis. The chemical genetics approach outlined in our study, using the analog-sensitive variant of a kinase, could be applied to assess the functional redundancy of other mitotic kinases and their impact on the phospho-regulation of PP4R3β.

Although CDK5-p25 complex activates CDK5 kinase activity for the phosphorylation of PP4R3β S840 in vitro (Fig. 3a), we did not observe CDK5 interaction with p35 or p25 in non-neuronal cells under condition of high CDK5 activity (Supplementary Fig. 5a). These results suggest that CDK5 activity in mitotic non-neuronal cells could be activated through an alternative mechanism. We found KIAA0528 as a putative CDK5 co-factor that is required for 53BP1 recruitment to DNA damage (Supplementary Fig. 5c, d). Unlike p35 however, KIAA0528 does not harbor a cyclin-like motif, and the precise mechanism by which KIAA0528 influences CDK5 activity needs to be explored. It is possible that alternative co-factor(s) might activate CDK5, the identification of which is beyond the scope of this study.

The function and substrates of CDK5 are well-studied in post-mitotic neurons; however, the role of CDK5 in cancer is not very well understood. Although CDK5 activity has been correlated with specific signaling pathways[37–40], currently there is no established substrate. We unequivocally demonstrate that PP4R3β as a bona fide substrate of CDK5 in non-neuronal cells. Future studies using quantitative phosphoproteomics, combined with chemical genetics that selectively inhibit CDK5 activity, will systematically identify other CDK5 substrates, and thus better elucidate the function of CDK5 in non-neuronal cells.

## Methods
**Cell lines, culture conditions, and transfection**. U2OS, HeLa, 293T, A375, and A375-CDK5 AS cells were cultured in DMEM-high glucose, pyruvate (Invitrogen, 11995-065) supplemented with 10% (v/v) fetal bovine serum (FBS) and 1% Pen-Strep (Invitrogen, 15140-122). hTERT-RPE1 cells were cultured in DMEM-F12 medium (Invitrogen, 21041-065) containing 10% FBS and 1% Pen-Strep. UWB1.289 cells were cultured in 1:1 MEBM (Lonza, CC-3151) and RPMI supplemented with 3% FBS, 1% Pen-Strep, MEGM SingleQuots (Lonza, CC-4136) containing BPE, hydrocortisone, hEGF, and insulin. Cells were maintained at 37 C in a 5% CO2 atmosphere with 21% oxygen. Transfection of plasmids was performed using Lipofectamine LTX reagent (Invitrogen, 15338-100) as per manufacturer's instructions. Transfection of siRNA was performed using RNAiMax (Invitrogen, 13778-150) as per manufacturer's instructions.

**siRNA sequences**. Sequences of siRNAs used in this study are as follows:
PP4R3β ORF siRNA 1: 5′-CCAUCUAUAUUGCGUAGUA-3′;
PP4R3β ORF siRNA 2: 5′-CACUUUCUUUGAAUCAUCC-3′;
CDK5 ORF siRNA 1: 5′-UUGCGGCUAUGACAGAAUC-3′;
CDK5 ORF siRNA 2: 5′-GAUGUCGAUGACCAGUUGA-3′
FIBP ORF siRNA: 5′-UAUGCAGCCAUCGUCUUCU-3′
KIAA0528 ORF siRNA: 5′-AACUCGGAGUGGUUUAAAU-3′

**Plasmids construction and generation of stable cell lines**. PP4R3β full length, fragments harboring residues 1–400, 1–700, and 721–849 were subcloned into pcDNA3.1-HisA-6xMyc using standard PCR and cloning methods. PP4R3β full length was also cloned into pOZ FLAG-HA. PP4R3β serine 840 phospho-variants, S840A, S840D, and S840F were generated using QuikChange II XL Site-Directed Mutagenesis Kit (Agilent, 200521) according to manufacturer's instructions. SiRNA-resistant constructs against PP4R3β ORF siRNA #1 were generated in the pcDNA3.1-HisA-6xmyc mammalian expression vector by site-directed mutagenesis. Primers used for the subcloning and site-directed mutagenesis reactions are listed in Supplementary Table 1. Construct expression frames and mutations were further verified by Sanger sequencing.

Analog-sensitive (AS) mutation phenylalanine 80 to glycine (F80G) in CDK5 was generated by QuikChange mutagenesis in p3X-FLAG CMV10 harboring CDK5. p3X-FLAG CMV10 CDK5-F80G was then transfected into A375 cells using Lipofectamine, with subsequent selection in 800 μg/ml of neomycin. The endogenous CDK5 in A375 cells was knocked out by CRISPR using lentiviral vector harboring the following guides:
Forward: 5′-CACCGCCGGGAGACTCATGAGATCG-3′;
Reverse: 5′-AAACCGATCTCATGAGTCTCCCGGC-3′.
The sequence of CDK5 that CRISPR guides target is CCGGGAGACTCATGAGATCG-(TGG < = NGG). Stable polyclonal cells with deletion of endogenous CDK5 were selected with Puromycin and verified by immunoblot.

**Cell cycle synchronization**. For synchronization to G2/M, cells were treated with RO-3306 (Calbiochem, 217699) dissolved in DMSO at the final concentration of 9 μM for 16–18 h. For synchronization to mitotic prometaphase, cells were treated with nocodazole (Sigma Aldrich, M1404) dissolved in DMSO at the final concentration of 100 ng/ml for 6 h.

**Cell cycle analysis**. Cells were fixed in cold 70% ethanol, processed with reagents from the FlowCellect Histone H2A.X Phosphorylation Assay Kit (EMD Millipore, FCCS100182) per manufacturer's instructions, hybridized with Anti-phospho Histone H3 (Ser10) antibody-Alexa Fluor 488 conjugate (EMD Millipore, 06–570-AF488), and stained with propidium iodide (Sigma-Aldrich, P4170). 10,000 cells per sample were analyzed on Beckman Coulter Cytoflex. Further data analysis was done using FlowJo software.

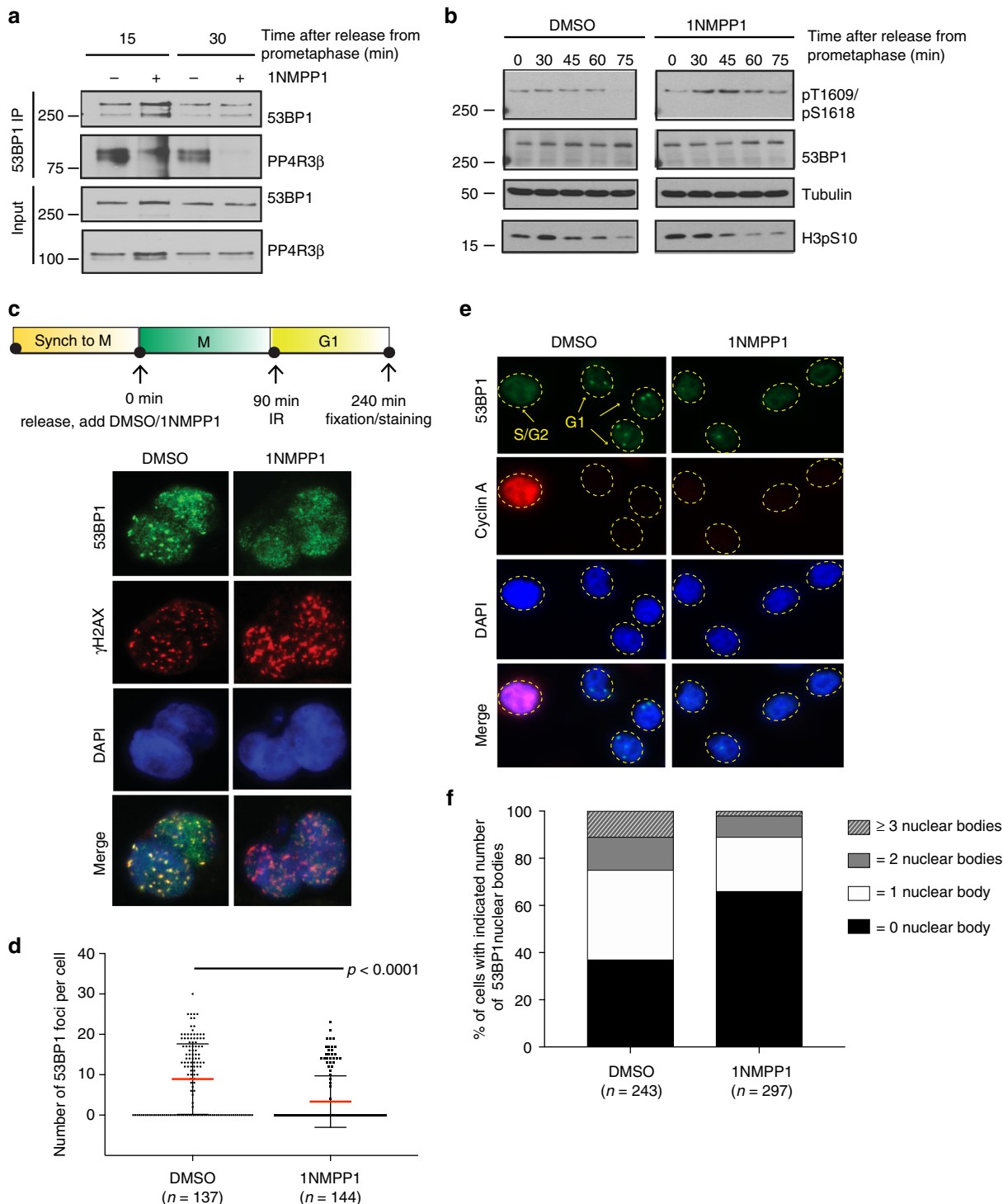

**Fig. 4** CDK5 regulates dephosphorylation and the recruitment of 53BP1 to DNA damage. **a** Endogenous 53BP1 was immunoprecipitated from DMSO-treated or 1NMPP1-treated A375-AS cells and collected at indicated time points after release from arrest at prometaphase. Levels of co-immunoprecipitated endogenous PP4R3β were analyzed by immunoblot. **b** Endogenous 53BP1 was immunoprecipitated from DMSO-treated or 1NMPP1-treated A375-AS cells, collected at indicated time points, after release from nocodazole-induced arrest at prometaphase. Levels of 53BP1 T1609/s1618 phosphorylation from immunoprecipitated endogenous 53BP1 were analyzed by immunoblot. **c** Mitotic A375-AS cells, collected by shake-off from arrest at prometaphase, were immediately treated with DMSO or 10 µM of 1NMPP1 and seeded on poly-D-lysine-coated coverslips. 90 min after release, treated cells were irradiated with 5 Gy. 240 min after release, cells were fixed and stained with indicated antibodies to detect irradiation-induced 53BP1 foci. γ-H2AX was stained to mark sites of DNA damage. **d** Quantifications of 53BP1 foci in **c**, pooled from triplicate repeat experiments, expressed as mean ± s.d. Total number of cells is indicated in parenthesis. P-values, Mann–Whitney U Test. **e** A375-AS cells were exposed to 0.2 µM Aphidicolin and DMSO or 10 µM 1NMPP1 for 12 h. Cells were fixed and stained for 53BP1 and Cyclin-A, a negative marker for G1 cells. **f** Quantification of percentage of cells containing 53BP1-containing nuclear bodies in **e** out of total cells, indicated in parenthesis, pooled from triplicate repeat experiments. P-values, Mann–Whitney U Test

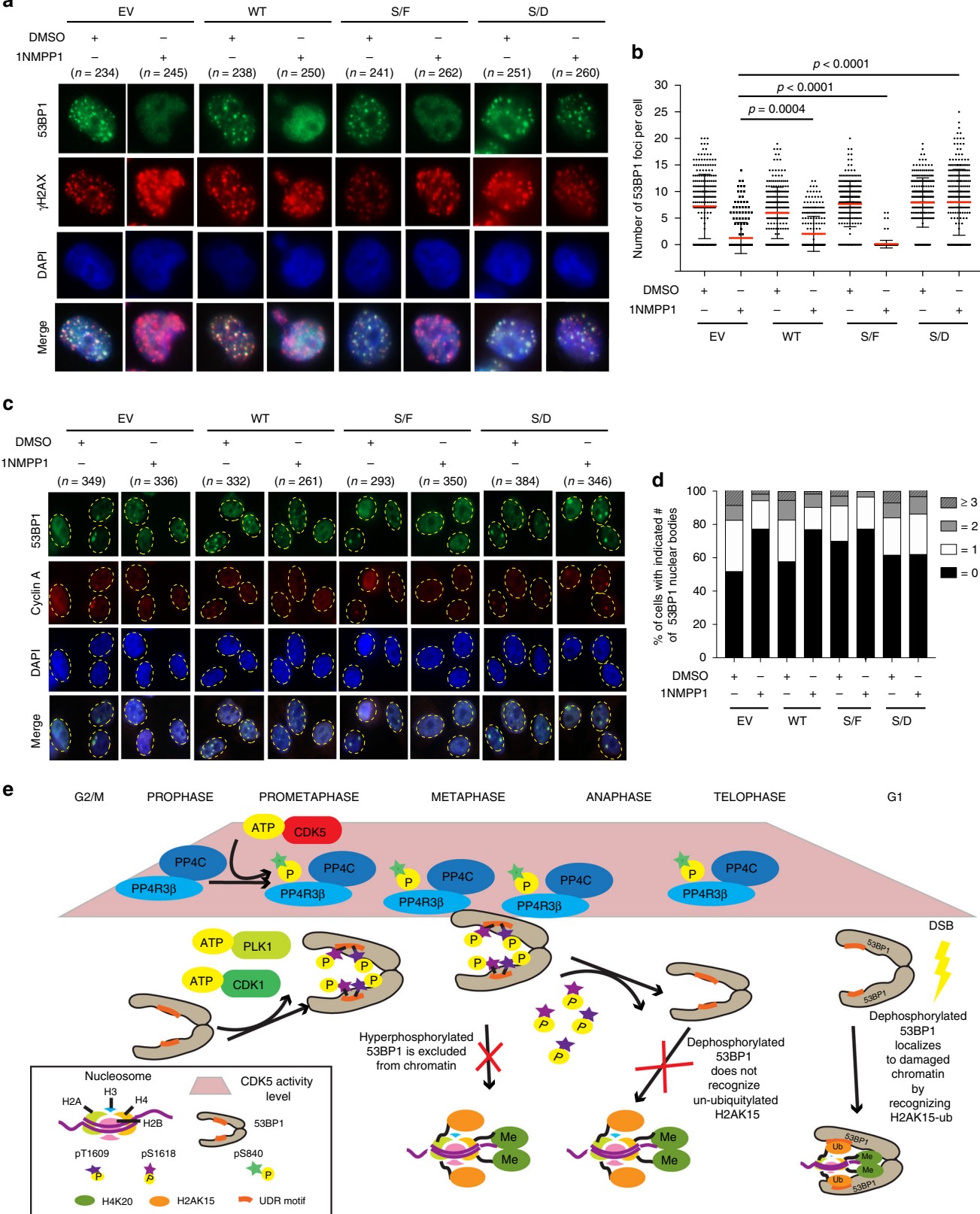

**Fig. 5** A CDK5-PP4 signaling axis drives 53BP1 recruitment to damaged chromatin. **a** A375-AS cells were depleted of endogenous PP4R3β by siRNA knockdown and transfected with indicated siRNA-resistant Myc-PP4R3β S840 phospho-variant constructs. Transfected cells were synchronized to prometaphase. Mitotic cells collected by shake-off were immediately treated with DMSO or 10 μM of 1NMPP1 and seeded on poly-D-lysine–coated coverslips. Ninety minutes after release, treated cells were irradiated with 5 Gy. 240 min after release, cells were fixed and stained with indicated antibodies to detect irradiation-induced 53BP1 foci. γ-H2AX was stained to mark sites of DNA damage. **b** Quantifications of 53BP1 foci in **a**, pooled from triplicate repeat experiments, expressed as mean ± s.d. Total number of cells is indicated in parenthesis. *P*-values, Mann–Whitney *U* Test. **c** A375-AS cells were depleted of endogenous PP4R3β by siRNA knockdown and transfected with indicated siRNA-resistant Myc-PP4R3β S840 phospho-variant constructs. The transfected cells were exposed to 0.2 μM Aphidicolin and DMSO or 10 μM 1NMPP1 for 12 h. Cells were fixed and stained for 53BP1 and Cyclin-A, a negative marker for G1 cells. **d** Quantification of percent cells containing 53BP1-containing nuclear bodies in **c** out of total cells (indicated in parenthesis) pooled from triplicate repeat experiments. *P*-values, Mann–Whitney *U* Test. **e** Model depicting CDK5 as a regulator of 53BP1 recruitment to damaged chromatin

**Chemicals**. Chemicals used in this study include 1NMPP1 (PP1 analog II, EMD Millipore, 529581); Olaparib (AZD2281/Ku-0059436; ChemieTek, CT-A2281); and aphidicolin (EMD Millipore, 178273).

**Protein extraction and immunoblotting**. Protein extracts were prepared in NETN buffer (50 mM Tris 7.5, 1 mM EDTA, 0.5% NP-40, 5% glycerol, 150 mM NaCl, cOmplete™ EDTA-free protease inhibitor cocktail (Sigma, 1183617001), and PhosSTOP™ phosphatase inhibitor cocktail (Sigma, 04906837001)). Briefly, cells were harvested and washed once in ice cold PBS before resuspending in lysis buffer. After 20 min incubation on ice, the cell lysate was clarified by centrifuging at 13,000 rpm for 10 min. Protein concentration in clarified lysate was determined by Pierce BCA Protein Assay Kit (Thermo Scientific, 23225) with reference to a standard curve generated with BSA. Extracts were mixed with 4X Laemmli sample buffer and heated at 95 °C for 5 min.

Denatured extracts were resolved on pre-cast NuPAGE™ 4–12%, 1.5 mm Bis-Tris polyacrylamide gels (Life Technologies, NP0335) and transferred to 0.2 μm nitrocellulose membranes (BioRad, 1620112). Membranes were blocked in 5% BSA (Sigma, 10735086001) in TBS-1% Tween20 for 1 h and incubated overnight in primary antibodies. Membranes were washed three times with TBS-0.05% Tween20 and probed with HRP-conjugated goat anti-mouse (Jackson ImmunoResearch, 115035008) or goat anti-rabbit (Jackson ImmunoResearch, 111035008) for 1 h (1:3000). Amersham ECL Western Blotting Detection Reagent Kit (GE Healthcare, RPN2106) was applied to develop the blots. Details of the primary antibodies used are provided in Supplementary Table 2.

**Antibodies**. Rabbit polyclonal anti-pT1609/S1618-53BP1 antibody was produced by Antagene (Sunnyvale, CA) with a peptide, Cys-NRLREQYGLGPYEAV(p)TPLTKAADI(p)SLDN. Rabbit polyclonal anti-pS840 antibody was produced by GL Biochem Ltd (Shanghai, China) with a peptide Cys-DEEEES-(p)S-PRKRPR. All other antibodies used in this study is listed in Supplementary Table 2.

**Immunofluorescence**. Cells were plated on coverslips, harvested at time points and fixed for 15 min with 4% (v/v) paraformaldehyde. Fixed cells were permeabilized for 5 min with 0.5% (v/v) Triton X-100 in PBS, washed and blocked in blocking solution (3% BSA in PBS) for 1 h at room temperature (RT). Cells were incubated with primary antibodies diluted in blocking solution for 2 h at RT. Following primary antibody incubation, cells were washed with PBS-0.05% Triton X-100 and incubated with respective secondary antibodies diluted in blocking buffer for 1 h at RT. Cells were washed before being mounted using DapiFuoromount-G (SouthernBiotech, 010020). Cells were imaged under a Zeiss Axio Imager using a ×63/1.40 oil objective lens. Mean value is represented in the scattered dot plots and analysis of statistical significance was performed using a Mann–Whitney test (GraphPad Prism).

For imaging of the phosphodynamics of 53BP1 T1609/S1618, cells were pre-extracted and fixed with 4% (v/v) PFA in CSK buffer (10 mM PIPES pH 6.8, 100 mM NaCl, 1 mM MgCl2, 1 mM EGTA and 0.5% Triton X-100) for 15 min. The cells were then fixed and stained as described above. Confocal microscopy for imaging of the phosphorylation dynamics of phosphorylated PP4R3β at S840 and that of phosphorylated 53BP1 at T1609 and S1618 was performed using a Zeiss Axiovert 200 M inverted microscope with a ×63 Plan Apo 1.4 NA objective. A series of nine 0.5 um optical sections were acquired with an ImagEM EMCCD camera (Hamamatsu) or a Prime BSI back-thinned sCMOS camera (Photometrics). Image acquisition parameters, shutters, filter positions and focus were set using Metamorph software. Images were analyzed using ImageJ software. Images presented in figures are maximum intensity projections of entire z-stacks. All antibodies used for immunofluorescence studies described herein are listed in Supplementary Table 2.

**Immunoprecipitation**. Cells were lysed in NETN lysis buffer. Cell lysates were incubated with antibody at 4 °C for 2 h and then incubated with Protein A/G Plus Agarose beads (Santa Cruz), or with antibody-conjugated matrix for 12–16 h. The immunoprecipitants was washed four times with NETN lysis buffer. The proteins were eluted from the beads in 60 μl of 2X Laemmli buffer with vigorous vortex for 1 min, follow by heating at 95 °C for 5 min. Eluates were resolved by SDS-PAGE. All antibodies used for immunoprecipitation studies described herein are listed in Supplementary Table 2.

**Mass spectrometry to identify PP4R3β phospho-S840**. HeLa cells stably expressing FLAG-HA PP4R3β were arrested in prometaphase with nocodazole treatment for 6 h. Mitotic cells were collected by shake off. Cells were lysed as described above. FLAG-HA PP4R3β was purified by tandem affinity purification using FLAG agarose beads and HA agarose beads and eluted by FLAG peptides and HA peptides, respectively. Immunoprecipitated proteins were diluted 1:1 with 100 mM ammonium bicarbonate, denatured with 0.1% RapiGest (Waters Corp., Milford, MA), reduced with 10 mM DTT (30 min 56 °C), alkylated with 22.5 mM iodoacetamide (30 min), and digested with trypsin overnight at 37 °C. After digestion, RapiGest was cleaved by acidifying the solution with 10% trifluoroacetic acid and incubating the mixture for 30 min at 37 °C. After centrifugation, peptides were desalted by C18 and phosphopeptides were enriched by $Fe^{+3}$-NTA-IMAC. The IMAC eluate was acidified and analyzed by nanoflow LC-MS/MS as described[41]. Raw data was converted to mgf using multiplierz scripts[42]. Database search was performed with Mascot version 2.2 against a human NCBI refseq database. Search parameters specified fixed carbamidomethylation of cysteine residues, as well as variable phosphorylation of serine, threonine, and tyrosine and oxidation of methionine. Precursor and product ion mass tolerances were 10 ppm and 0.6 Da. Phosphosite localization was performed using mzStudio[43]. Phosphosite assignment and localization was verified by analysis of MS/MS spectra acquired by direct infusion for the synthetic peptide H-GSLVGLVDYPDDEEEDEEEESS$^{phos}$PR-OH.

**Protein expression and purification**. His-tagged CDK5 and p25 cloned into the pFastBac vector were expressed using the Bac-to-Bac baculovirus expression system. Cell pellets of each protein were combined and lysed in Tris buffer (pH 8.0) containing 150 mM NaCl and 1% NP-40 and batch purified using 100 μl of Ni-NTA beads. PP4R3β S840 wild-type and S840A variant were expressed in BL21 DE3 E. coli cells. Cells were grown at 37 °C to an OD600 of ~0.5 and expression was induced with 500 μM IPTG followed by overnight growth at room temperature. Frozen pellets were lysed in Tris buffer (pH 8.0) containing 500 mM NaCl, 10% glycerol, and 1% NP-40 via French press and batch purified using 100 μl Ni-NTA beads. Protein levels were normalized using Coomassie staining prior to the kinase assay.

**In vitro kinase assays**. For in vitro labeling, Cdk5-p25 complex (on Ni-NTA beads) was pre-incubated with 100 μM of ATP for 1 h in a 1× kinase buffer (50 mM Tris, 10 mM MgCl2) to reduce background phosphorylation. The beads were washed with 1× kinase buffer 3 times to remove excess ATP, eluted and then subjected to an in vitro kinase assay with 2 mg of 6×-His-tagged recombinant protein (wild-type or mutant S840A PP4R3β) in the presence of 0.5 mCi of [γ-32P] ATP for 15 min. Reactions were terminated upon the addition of SDS loading buffer and subsequently separated by SDS-PAGE gel, transferred to PVDF membrane and exposed for autoradiography. To measure kinase activity kinetics during mitosis, cells released from RO-3306-induced G2/M arrest or nocodazole-induced prometaphase arrest were lysed in 1% NP-40 lysis buffer (1% NP-40, 20 mM Tris, pH 8.0, 150 mM NaCl, 1 mM PMSF, 10 μg/ml leupeptin, and 10 μg/ml aprotinin and cleared by centrifugation at 4 °C. Cleared lysates were mixed with CDK1 or CDK5 antibody and protein A Sepharose beads and incubated at 4 °C for 2 h. Immune complexes were washed twice with 1% NP-40 lysis buffer and twice with kinase buffer (50 mM Tris, pH 8.0, 20 mM MgCl2). Immune complexes were subjected to in vitro kinase assays using [32P]-ATP and 5 μg of CDK1/ CDK5 substrate peptide (KHHKSPKHR) in a final volume of 30 μl. After 20 min, the reactions were terminated by spotting 25 μl of the reaction volume onto p81 phosphocellulose disks (Whatman, Clifton, NJ) and immersing in 100 ml of 10% acetic acid for 30 min, followed by three washings in 0.5% phosphoric acid (5 min each) and finally rinsing with acetone. The radioactivity was measured in a liquid scintillation counter.

**Viability assays**. For clonogenic assays, 500,000 HeLa cells were seeded per well in 6-well plates on first day. On day 2, the seeded cells were transfected with 3 μg of siRNA-resistant pCDNA-PP4R3β expressing wild-type or indicated S840 phosphovariants. On day 3, the transfected cells were trypsinized and reseeded into a 60 mm plate, and reverse-transfected with 100 pmol of PP4R3β-targeting siRNA. On day 4, treated cells were trypsinized and counted. 600 cells were plated per well in 6 well plates. On day 5, cells were irradiated at indicated doses and allowed to form colonies for 10 days before being stained by 0.1% crystal violet solution. Surviving colonies greater than 1 mm diameter were counted.

For colorimetric assays, UWB1.289 cells were transfected with PP4R3β-expressing constructs and siRNA as described above for clonogenic assays. On day 4, 1000 transfected cells were seeded per well into a 96-well plate. On day 5, cells were treated with indicated dosage of Olaparib for 96 h before being quantified by Cell-Titer-Glo Luminescent Cell Viability Assay per manufacturer's instructions (Promega, G7571).

**Quantification and statistical analysis**. Prism 7 (GraphPad Software Inc.) was used for statistical analyses and production of all graphs and plots. Unless mentioned otherwise, all statistics were evaluated by Mann–Whitney $U$ Test.

**Reporting summary**. Further information on research design is available in the Nature Research Reporting Summary linked to this article.

## Data availability

The source data underlying Figs. 1b–g, 2b, d–g, 3b, d–h, 4a, b, d, f, 5a, b; and Supplementary Figs. 1a, i, 2b–f, 3b, e, f, h, 4c, e–g and 5a, b, d are provided as a Source Data file. All other original data that support the findings of this study are available from the corresponding author upon reasonable request.

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

## Acknowledgements

D.C. is supported by R01 CA208244 and R01CA142698, Leukemia and Lymphoma Society Scholar Grant, DOD Ovarian Cancer Award W81XWH-15-0564/OC140632, and the Claudia Adams Barr Program in Innovative Basic Cancer Research. X.-F.Z. is supported by Postdoctoral Fellowship PF-14-181-01-DMC from the American Cancer

Society. K.N.C. is supported by Postdoctoral Fellowship PF-18-115 from the American Cancer Society. P.S. is supported by NIH grants R01CA239660 and R01CA236226. J.A.M. is supported by NIH grants P01 CA203655, R01 CA215489, R01 NS050674, and the Dana-Farber Strategic Research Initiative. K.S. is supported by R01 CA208244, NSF Award # 1708823 and U.S. Army Medical Research Acquisition Activity, Prostate Cancer Research Program Award # PC130391. A.N. is supported by P50 CA127297.

## Author contributions

X-F.Z. designed, performed, and analyzed the majority of the experiments in this study, with assistance from S.S.A. K.N.C. acquired images and performed quantitative analysis to monitor the PP4R3β and 53BP1 phosphorylation kinetics during mitosis. G.A. performed mass spectrometric analyses. K.V. performed the CDK5 in vitro kinase assay. K.N. and S.R.S. measured CDK5 and CDK1 activity in mitotic cell extract. S.S. generated the A375 cells expressing the CDK5 analog-sensitive variant. S.R. and A.N. synthesized the CDK5 inhibitor 20-223. X-F.Z. and D.C. conceived the study and wrote the paper, with input from A.N., P.S., J.A.M., and K.S.

## Additional information

**Competing interests:** J.A.M. serves on the Scientific Advisory Board of 908 Devices. P.S. has been a consultant at Novartis, Genovis, Guidepoint, The Planning Shop, ORIC Pharmaceuticals and Exo Therapeutics; his laboratory receives research funding from Novartis. The remaining authors declare no competing interests.

