## [Peer Review File · Nature Communications]

Reviewers' comments:

Reviewer #1 (Remarks to the Author):

This report is the extension of the previous finding from the same group showing that 53BP1 is phosphorylated on two residues, T1609, and S1618 during mitosis to prevent the binding of 53BP1 to DNA. They also showed that PP4C/PP4R3 β complex dephosphorylates these residues to allow the recruitment of 53BP1 to chromatin in early G1 phase (Mol Cell. 2014, 54:3:512-25). In the current manuscript, they reported the novel upstream regulators of PP4C/PP4R3 β to control this signaling. Surprisingly, CDK5, known to mainly function in neurons, has a novel role in regulating 53BP1 in non-neuronal cells. During the late-mitotic phase, CDK5 phosphorylates PP4R3 β at S840, which is required for PP4C/PP4R3 β complex to dephosphorylate 53BP1, leading to the recruitment of 53BP1 to DNA. It has been known that DNA damage response is attenuated in cells undergoing mitosis to keep genomic stability, however underlying mechanisms are still unclear. The novel mechanism proposed in this study is interesting and will have a broad and significant interest in the field of cancer, DNA damage, and mitosis in general. I have several concerns that should be addressed before the publication.

Major Concern

My major concern is the timing between CDK5 activity, S840 phosphorylation of PP4R3 β , and T1609 and S1618 phosphorylation of 53BP1. Presented data are not always consistent with the model proposed in 5e. Based on Figure 1e, 53BP1 is phosphorylated in early M phase (0h), late M phase (1h), and early G1 (3h) in control cells and PP4R3 β knockdown cells expressing wild-type PP4R3 β , suggesting that PP4C/PP4R3 β complex is inactive in this period, and therefore, 53BP1 cannot bind DNA. Thus, I expect low CDK activity and low S840 phosphorylation of PP4R3 β in M phase and early G1. PP4C/PP4R3 β complex is supposed to be activated by PP4R3 β phosphorylation around or after early G1. However, Figure 2a and 2b showed that induction of S840 phosphorylation already occurs even in prophase (15min, early M phase) while Figure 2d shows that CDK5 activity becomes highest around telophase (120 min, late M1). Figure 3g is even more confusing. Control cells showed that S840 phosphorylation of PP4R3 β is high in the middle M phase, and S840 phosphorylation disappears in late M phase and early G1, which is inconsistent with data showing in Figure 1e and 2ab. Figure 2ab and S2bc are also confusing. While 53BP1 dephosphorylation occurs in Anaphase in Figure S2bc, induction of PP4R3 β phosphorylation already occurs in prophase in Figure 2ab. Based on the model in Figure 5e, the transition of 53BP1 from phosphorylated form to non-phosphorylated form by PP4R3 β phosphorylation occurs in the late M phase.

This inconsistency could be derived from using different cell lines in each experiment. They used HeLa cells in Figure 1e and 2d, RPE1 cells in Figure 2ab and S2bc, and A375 cells in Figure 3g. However, I expect that the mechanism of attenuation of DNA damage response in M phase should be fundamental and similarly conserved in many human cell lines. To avoid confusion, authors must monitor the precise time course of phosphorylation events of PP4R3 β and 53BP1 together by "western blotting" during G2/M, early, mid, late M phase and early and mid G1 at least two cell lines since the authors have good antibodies to detect them. There is no figure showing phosphorylation of PP4R3 β and 53BP1 together by western blotting, which will be critical for this manuscript. It is also good to show these results accompanied by the immunostaining and CDK5 activity as shown in Figure 2a, S2b, and 2d.

Minor Concerns

Figure 1e: It is not clear about the extent of overexpression of PP4R3 β wild-type and S/F mutant compared to the expression of endogenous PP4R3 β . To avoid artifact of too much overexpression,

authors should compare the levels of ectopic PP4R3 β wild-type and S/F mutant compared to endogenous PP4R3 β using PP4R3 β antibody.

Figure S3a: knockdown of PP4R3 β should also be included to test the antibody specificity.

Figure 2d: CDK5 activity in the later point such as the early G1 phase should be provided since the early G1 phase is also an important phase to compare with M phase as shown in their model in figure 5e.

Figure 2e was first mentioned after explaining Figure 4 in the main text. It is a bit confusing to read.

In the discussion, they mentioned that “keeping 53BP1 ‘inactive’ in mitosis is necessary for genomic stability”. As I mentioned in the major concern, some figures (like 1e) show 53BP1 is inactive in M phase and early G1, but some figures (like S2b) show 53BP1 is only transiently inactive in Prometaphase and Metaphase. This sentence should be modified depending on the results in the revision I requested. The model figure in Figure 5e is also drawn more precisely depending on the results in the revision.

In the abstract, the authors mentioned that “CDK5 is active in mitotic non-neuronal cells”. This will prompt readers to think that CDK5-PP4C/PP4R3 β -mediated 53BP1 activation occurs in all the M phase, which is inconsistent with the idea that “keeping 53BP1 ‘inactive’ in mitosis. In fact, CDK5 is only getting active in metaphase in Figure 4d. So the sentence should be modified to avoid confusion.

Reviewer #2 (Remarks to the Author):

Nature Communications Review: “A CDK5-PP4 phospho-signaling cascade drives 53BP1 to sites of DNA damage”

In this paper, the Chowdhury group investigates the regulation of the PP4R3 β -53BP1 interaction and its role in controlling 53BP1 localization to sites of DNA damage in G1. Previous work published from this lab (Lee et al., *Molecular Cell*, 2014) characterized the specific residues of 53BP1 that are de-phosphorylated by PP4R3 β but the mechanism by which PP4R3 β recognizes 53BP1 remained unknown. In this study, the authors find that phosphorylation of PP4R3 β at S840 mediates interaction with 53BP1 and that mutation at this site impairs 53BP1 localization to IR-induced foci. Through prediction analysis and experimental approaches, the authors find that CDK5, a non-cyclin kinase, phosphorylates serine 840 of PP4R3 β to mediate its interaction with 53BP1. The authors further explore pharmacological inhibition of CDK5 to establish its importance in promoting 53BP1 recruitment to DNA damage-induced foci.

Overall, the data presented in this manuscript is of good quality and mostly convincing. How 53BP1 recruitment to DNA lesions is a relevant question that is not yet fully understood. The proposed mechanism by which CDK5 phosphorylates PP4R3 β to mediate the PP4R3 β -53BP1 interaction and subsequent 53BP1 dephosphorylation is elegant and well explained. One concern about this manuscript is that many of the experiments are dependent on cell cycle synchronization and further added controls would strengthen the authors' conclusions. In addition, the text needs to be improved as it contains many typos and unclear sentences. Below are specific concerns/suggestions that I feel should be addressed before this paper is considered for publication.

- 1) Does inhibition of CDK5 cause sensitivity to IR in WT cells or increase resistance of brca1-KO cells to PARP inhibitors?
- 2) For Figure 2a, it would be useful to show the dynamics of T1609/S1618 phosphorylation (Supp. Fig. 2bc) just next to the dynamics of S840 phosphorylation (Fig. 2ab). I suggest that Supp. Fig. 2bc gets moved to main figure 2. Also, since this is a key part of the paper, it would be nice it is confirmed using an independent approach, such as by western blot.
- 3) Results from Figure 3g (inhibition of S80 upon CDK5 inhibition) are also key for the paper, but the presented WB is weak and lacks quantification and statistics.
- 4) The beginning of the results section is confusing: "Phosphorylation of PP4R3a specifically regulates its interaction with 53BP1. PP4 regulatory subunits PP4R3a and PP4R3 β share 67% homology, but PP4R3a diverges significantly from PP4R3 β in its C-terminal region (aa 721-849)9. We hence hypothesized that the PP4R3 β C-terminal region mediates its specific interaction with 53BP1."
 - a. Does it mean that PP4R3a does not bind (the WB shows otherwise)? Please clarify what is the logic of this sentence.
- 5) Several experiments presented in this manuscript are predicated on cell cycle synchronization, which is highly variable in different treatment conditions and between cell lines. The authors should verify that they have successfully synchronized cells using a flow-cytometry based assay for the following experiments: figure 1C/D, 1F and 1E (although E does have H3pS10 as a control, which is appreciated), 2a, 2d, 3c-g, 4c-f.
- 6) For experiments using nocodazole arrest and release protocol, the authors mention that released cells are in G1. How do they know that in that time-frame cells have not entered net S-phase?
- 7) Supplemental Figs 1e and 1f (sensitivity to IR and resistance to PRPi) should be added to main figure. This is an important results.
- 8) It is somewhat confusing to understand what cell lines and treatment conditions are used in each experiment, and some of this information is also absent from the main text. To improve clarity, it would be helpful to indicate which cell line and/or drug treatment is used in each figure itself in addition to the figure legend. For example, figure 1A should clearly indicate that 293T cells are used whereas figure 1C should indicate the samples were harvested from nocodazole-arrested HeLa cells.
- 9) An empty vector control is missing from figure 1D
- 10) Figure 1F: DAPI images in the WT and SF panel have been switched
- 11) An asynchronous control for figure 1C and D would be something to consider adding, as it would support the idea that these interactions specifically enriched in the synchronized population
- 12) Typo in figure legend 2, there's a bold (a) where there shouldn't be.
- 13) The final model should indicate important residues identified in this study, such as S840.
- 14) Figure 2c and 2e could be moved to supplemental
- 15) For figure 2a/b, it could be interesting to see co-localization and staining of other factors such as 53BP1 or other markers of DNA repair to verify the localization pattern.
- 16) The text needs to be improved as it contains many typos and sentences that are unclear.

Reviewers' comments:

Reviewer #1 (Remarks to the Author):

1) Major Concern

My major concern is the timing between CDK5 activity, S840 phosphorylation of PP4R3 β , and T1609 and S1618 phosphorylation of 53BP1.

This inconsistency could be derived from using different cell lines in each experiment. They used HeLa cells in Figure 1e and 2d, RPE1 cells in Figure 2ab and S2bc, and A375 cells in Figure 3g. However, I expect that the mechanism of attenuation of DNA damage response in M phase should be fundamental and similarly conserved in many human cell lines. To avoid confusion, authors must monitor the precise time course of phosphorylation events of PP4R3 β and 53BP1 together by “western blotting” during G2/M, early, mid, late M phase and early and mid G1 at least two cell lines since the authors have good antibodies to detect them. There is no figure showing phosphorylation of PP4R3 β and 53BP1 together by western blotting, which will be critical for this manuscript. It is also good to show these results accompanied by the immunostaining and CDK5 activity as shown in Figure 2a, S2b, and 2d.

We thank the reviewer for these insightful comments and suggestions. Indeed, there are significant differences among the cell lines in the response to synchronization (HeLa cells respond to RO-3306 synchronization to G2/M and Nocodazole to prometaphase while A375 cells only respond to nocodazole), and the duration of the mitosis (HeLa cells require 5 hours post release from prometaphase to reach G1 while A375 only requires 75 minutes). Thus, the kinetics and time scale of the phosphorylation events are indeed cell-line dependent. Based on reviewer's suggestions, we have now conducted immunoblotting for the phosphorylation events in PP4R3 β and 53BP1, and CDK5 kinase activity measurement during G2/M, different phases of mitosis and early/mid G1, with the appropriate time scale for two distinct cell lines (HeLa and A375).

In addition, accuracy in distinguishing stages of mitosis depends on the experimental approach. In the immunofluorescence experiment to measure PP4R3 β and 53BP1 phosphorylation across mitosis in RPE1 in updated Fig 2, we can visualize cells at distinct stages of mitosis, based on chromatin morphology shown by DAPI staining. Hence, we can definitively select a population of cells at a specific stage of mitosis and concordantly measure the phospho-staining intensity in those cells. For the immunoblotting and kinase activity measurement experiments, despite prior synchronization of cells at G2/M or prometaphase, bulk cell population collected at time points for measurement harbors cells at different stages of mitosis. This is evident in Figure 2e for HeLa cells post release from prometaphase and Supplementary Figure 4c for A375 post release from prometaphase. Hence, we approximately correlate the time of collection/assay for phospho-events and kinase activities to representative phases of mitosis.

Our results from immunoblotting of the phospho-status of PP4R3 β and 53BP1 and the CDK5 kinase activity in HeLa (Fig 2e-g) and A375 (Supp Fig 4 c-g) correlate with the microscopy of the immunostaining of phospho-PP4R3 β and phospho-53BP1 in RPE1 (Fig 2a-d) in terms of the

mitotic stage for the occurrence of phosphorylation and CDK5 kinase activities. Specifically, both CDK5 kinase activity and PP4R3 β phosphorylation correlate during mitosis, low levels during prophase, peaking mid-mitosis (metaphase/anaphase), continuing to remain high in late mitosis (anaphase/telophase) and declining as cells proceed to G1. 53BP1 T1609-S1618 are phosphorylated during prometaphase and metaphase (Fig 2a-b; Fig 1c/previously Fig 1e at 0 hr post-release from prometaphase in HeLa cells), by CDK1 and PLK1, and dephosphorylated during late stage of mitosis with rapid reduction around anaphase/telophase. These results continue to support our hypothesis that CDK5 mediated phosphorylation of PP4R3 β is necessary for the dephosphorylation of 53BP1.

We recognize the reviewer's point that PP4R3 β S840 phosphorylation is evident in prophase (Figure 2c-d), yet 53BP1 dephosphorylation does not occur until after metaphase. Although these are distinct phases of mitosis, but they are relatively short (15-20 min), and it is difficult to ascertain precisely how long it takes the phosphorylated PP4R3 β to recognize and capture phosphorylated 53BP1 in the subcellular milieu of a mitotic cell. We speculate that timing of PP4R3 β S840 phosphorylation and dephosphorylation of 53BP1 may not precisely match because of various factors including the possibility that additional factors/modifications facilitate the PP4R3 β mediated dephosphorylation of 53BP1. Also, it is very likely that 53BP1 is not the only substrate of PP4/PP4R3 β and PP4R3 β is not the only substrate of CDK5. Hence despite 53BP1 being dephosphorylated already after metaphase, CDK5 and PP4R3 β activities persist through late stages of mitosis for other substrates. We would like to emphasize that when 53BP1 T1609-S1618 becomes dephosphorylated in anaphase, there is high PP4R3 β S840 phosphorylation and high CDK5 activity in different cell lines and with different readouts (immunoblot, immunofluorescence, kinase assays). This observation strongly supports our hypothesis that CDK5-mediated phosphorylation of PP4R3 β at S840 is critical for the dephosphorylation of 53BP1 at T1609-S1618.

Minor Concerns

Figure 1e: It is not clear about the extent of overexpression of PP4R3 β wild-type and S/F mutant compared to the expression of endogenous PP4R3 β . To avoid artifact of too much overexpression, authors should compare the levels of ectopic PP4R3 β wild-type and S/F mutant compared to endogenous PP4R3 β using PP4R3 β antibody.

We appreciate reviewer's concern. To address this concern, we have included western blot panels in updated Supplementary Figure 1g. The top panel is a PP4R3 β blot that indicates both endogenous PP4R3 β and exogenously-expressed Myc-PP4R3 β constructs. The expression level of exogenous Myc-tagged WT and S/F mutant appear comparable to endogenous PP4R3 β , if not slightly less. The accompanying updated part in the Supplementary Figure Legend is highlighted in yellow.

Figure S3a: knockdown of PP4R3 β should also be included to test the antibody specificity.

We appreciate reviewer's suggestion. The original Supplementary 3a, b, c that demonstrated the working of the phospho-antibody for PP4R3 β are now combined with supplementary Figure 2a as Supplementary Figure 2 to demonstrate the working of phospho-antibodies for 53BP1 pT1609/pS1618 and for PP4R3 β pS840. The authors have now included Supplementary Fig 2d that demonstrated specification of PP4R3 β pS840 antibody, indicated by the lack of signal in cells knocked down of PP4R3 β . The accompanying updated part in the Supplementary Figure Legend is highlighted.

Figure 2d: CDK5 activity in the later point such as the early G1 phase should be provided since the early G1 phase is also an important phase to compare with M phase as shown in their model in figure 5e.

The authors remeasured CDK5 kinase activity across mitosis into G1 in HeLa cells, as indicated in Figure 2h, with representative accompanying cell cycle phase shown in Figure 2e and 2g. In HeLa cells, at 7 hours post release from RO-3306, >95% of the cells have reached G1, and CDK5 activity have decreased to a level close to early mitosis. Furthermore, in Figure 2a and 2b, the authors have redone the measurement of PP4R3 β S840 phosphorylation intensity to include cells at G1 stage, and observed that the phosphorylation intensity decreased in G1 to comparable level as that during G2/M. This result correlates with the CDK5 kinase activity throughout mitosis.

Figure 2e was first mentioned after explaining Figure 4 in the main text. It is a bit confusing to read.

We apologize for the confusion and have now revised the text accordingly.

In the discussion, they mentioned that "keeping 53BP1 'inactive' in mitosis is necessary for genomic stability". As I mentioned in the major concern, some figures (like 1e) show 53BP1 is inactive in M phase and early G1, but some figures (like S2b) show 53BP1 is only transiently inactive in Prometaphase and Metaphase. This sentence should be modified depending on the results in the revision I requested. The model figure in Figure 5e is also drawn more precisely depending on the results in the revision.

We acknowledge reviewer's confusion that arise from the incongruency between the timing of 53BP1 dephosphorylation that occurs as soon as after metaphase (as shown in initial S2b/updated figure 2c), and lack of 53BP1 recruitment to chromatin in late mitosis. We have indeed modified the Discussion section. We recognize that CDK5-PP4-mediated dephosphorylation of 53BP1 is *an* important mechanism of restoring 53BP1 activity, specifically in its ability to be localize to damaged chromatin, but by no means, the only step involved in 53BP1 recruitment to damaged chromatin. For example, Orthwein et al. 2014, Science (Durocher lab), showed that hyperphosphorylation of RNF8 prevents ubiquitylation H2AK15 in mitosis thereby restricting 53BP1 recruitment. Therefore, dephosphorylation of RNF8 sometime during mitosis is an additional step necessary to fully restore 53BP1 recruitment to damaged chromatin.

Detailed investigation of additional mechanism(s) to fully restore 53BP1 recruitment to damaged chromatin, however, is beyond the scope of the current study.

In the abstract, the authors mentioned that “CDK5 is active in mitotic non-neuronal cells”. This will prompt readers to think that CDK5-PP4C/PP4R3 β -mediated 53BP1 activation occurs in all the M phase, which is inconsistent with the idea that “keeping 53BP1 ‘inactive’ in mitosis. In fact, CDK5 is only getting active in metaphase in Figure 2d. So the sentence should be modified to avoid confusion.

We appreciate reviewer’s suggestion and recognize the inconsistency. The authors have modified the sentence in question accordingly, to “CDK5 is active during the mid to late mitotic phase in non-neuronal cells.”

Reviewer #2:

Overall, the data presented in this manuscript is of good quality and mostly convincing. How 53BP1 recruitment is regulated to DNA lesions is a relevant question that is not yet fully understood. The proposed mechanism by which CDK5 phosphorylates PP4R3 β to mediate the PP4R3 β -53BP1 interaction and subsequent 53BP1 dephosphorylation is elegant and well explained. One concern about this manuscript is that many of the experiments are dependent on cell cycle synchronization and further added controls would strengthen the authors’ conclusions. In addition, the text needs to be improved as it contains many typos and unclear sentences. Below are specific concerns/suggestions that I feel should be addressed before this paper is considered for publication.

1) Does inhibition of CDK5 cause sensitivity to IR in WT cells or increase resistance of brca1-KO cells to PARP inhibitors?

We appreciate reviewer’s suggestion and have included these data accordingly, in updated Figure 3e to show CDK5 inhibition by treatment with compound 20-223 confers sensitivity to irradiation; and in updated Figure 3f to show CDK5 inhibition by treatment with compound 20-223 confers resistance to PARP inhibitor olaparib. The accompanying updated part in the main Figure Legend is highlighted.

2) For Figure 2a, it would be useful to show the dynamics of T1609/S1618 phosphorylation (Supp. Fig. 2bc) just next to the dynamics of S840 phosphorylation (Fig. 2ab). I suggest that Supp. Fig. 2bc gets moved to main figure 2. Also, since this is a key part of the paper, it would be nice if it is confirmed using an independent approach, such as by western blot.

The authors have changed the figures accordingly. Supplementary figure 2b and c that showed 53BP1 T1609/S1618 phosphorylation intensity and quantification through mitosis are now moved to main figure 2 as Figure 2a and b.

Figure 2a and b show representative immunofluorescence images of individual RPE1 cell at selected mitotic stage, as indicated by chromatin state (DAPI stain), and concordant mean fluorescence intensity reflecting phosphorylation.

Phosphorylation of 53BP1 at T1609/S1618 by immunoblot (updated Fig 2g) is detected and shown from bulk cells corresponding to the approximate representative mitotic phases, as shown in updated Fig 2e.

3) Results from Figure 3g (inhibition of S840 upon CDK5 inhibition) are also key for the paper, but the presented WB is weak and lacks quantification and statistics.

The authors recognize reviewer's concern and have quantified the phospho-S840 signal in DMSO set and 1NMPP1 set relative to input PP4R3 β , from duplicate experiments, and presented quantification and statistics in updated Figure 3h. The accompanying updated part in the main Figure Legend is highlighted in yellow.

4) The beginning of the results section is confusing: "Phosphorylation of PP4R3 α specifically regulates its interaction with 53BP1. PP4 regulatory subunits PP4R3 α and PP4R3 β share 67% homology, but PP4R3 α diverges significantly from PP4R3 β in its C-terminal region (aa 721-849)9. We hence hypothesized that the PP4R3 β C-terminal region mediates its specific interaction with 53BP1."

a. Does it mean that PP4R3 α does not bind (the WB shows otherwise)? Please clarify what is the logic of this sentence.

We apologize for the confusion and have edited the manuscript to clarify this point. In our previous study (Lee et al. 2014 *Molecular Cell*) we have shown that only knockdown of PP4R3 β , and not PP4R3 α , impairs the dephosphorylation of 53BP1 at T1609/S1618. Therefore, we focused on the interaction between PP4R3 β and 53BP1.

5) Several experiments presented in this manuscript are predicated on cell cycle synchronization, which is highly variable in different treatment conditions and between cell lines. The authors should verify that they have successfully synchronized cells using a flow-cytometry based assay for the following experiments: figure 1C/D, 1F and 1E (although E does have H3pS10 as a control, which is appreciated), 2a, 2d, 3c-g, 4c-f.

The accompanying flow cytometry plots that include measurement for H3pS10 and propidium iodide staining are presented to verify the synchronization of the cells in the experiments that the reviewer listed. The flow cytometry accompaniment is listed in the table below:

Figures in the initial	Corresponding	Relevant flow cytometry figure in the revised
---------------	---

submitted manuscript	figure in the revised manuscript	manuscript; rationale for not including flow cytometry
Figure 1c-d	Figure 1b	Supplementary Figure 1f-h
Figure 1e	Figure 1c	Supplementary Figure 1h, 1j
Figure 1f-g	Figure 1d-e	Supplementary Figure 1j
Figure 2a-b	Figure 2c-d	In this immunofluorescence experiment, cells at distinct stages were selected based on chromatin/DAPI staining, hence PP4R3 β phosphorylation is not quantified from bulk cell population.
Figure 2d (RPE1)	Figure 2h (HeLa)	Supplementary Figure 1h (HeLa)
Figure 3c-d	Supplementary Figure 3d-e	Supplementary Figure 3g
Figure 3e-f	Figure 3c-d	Supplementary Figure 3i
Figure 3g	Figure 3g-h	Supplementary Figure 4d, h
Figure 4c-d	Figure 4c-d	Supplementary Figure 4h
Figure 4e-f	Figure 4e-f	Cells were not synchronized in this experiment. S/G2 cells, indicated by Cyclin A-positive staining, were excluded from quantification.

6) For experiments using nocodazole arrest and release protocol, the authors mention that released cells are in G1. How do they know that in that time-frame cells have not entered net S-phase?

As indicated in all of the flow cytometry experiments above, the cells released from G2/M and prometaphase are clearly in G1 phase and are analyzed no longer than 7 hours post release. It is known that G1 phase lasts 10-12 hours, hence the authors are confident that the synchronized cells that were analyzed were not in S phase state.

7) Supplemental Figs 1e and 1f (sensitivity to IR and resistance to PARPi) should be added to main figure. This is an important result.

We appreciate reviewer's suggestion and have revised accordingly. We have moved Supplemental Fig 1e and 1f to main figure now as Fig 1f and g; and for space and conceptual narrative purpose, moved previous Fig 1a to Supplemental Fig 1a. The accompanying updated part in the main Figure Legend is highlighted in yellow.

8) It is somewhat confusing to understand what cell lines and treatment conditions are used in each experiment, and some of this information is also absent from the main text. To improve clarity, it would be helpful to indicate which cell line and/or drug treatment is used in each figure itself in addition to the figure legend. For example, figure 1A should clearly indicate that 293T cells are used whereas figure 1C should indicate the samples were harvested from nocodazole-arrested HeLa cells.

We thank the reviewer for the suggestion and have now included the necessary information in the revised figure legend.

9) An empty vector control is missing from figure 1D.

We have included empty vector control for the IPs, specifically Myc-empty vector (EV) in place of PP4R3 β and FLAG empty vector (“-” sign) in place of 53BP1, as shown in updated Figure 1b.

10) Figure 1F: DAPI images in the WT and SF panel have been switched

We have corrected this oversight in the now updated Figure 1d (previously Figure 1F).

11) An asynchronous control for figure 1C and D would be something to consider adding, as it would support the idea that these interactions specifically enriched in the synchronized population

We have redone the IP experiment to include a set of asynchronous samples, now shown in updated Figure 1b. The IP results indicate an enriched interaction between 53BP1 and PP4R3 β in nocodazole-arrested mitotic cells.

12) Typo in figure legend 2, there's a bold (a) where there shouldn't be.

We corrected the oversight in the figure legend text.

13) The final model should indicate important residues identified in this study, such as S840.

We appreciate reviewer's suggestion and have now indicated PP4R3 β S840 phosphorylation in the figure with a green star and annotated accordingly in the figure key.

14) Figure 2c and 2e could be moved to supplemental

We appreciate reviewer's suggestion and have now moved Fig 2C to now Supplemental Fig 3a. We also moved Fig 2d, e (CDK5 kinase activity) to now Supplemental Fig 3 b, c, in combination with CDK1 kinase activity to highlight the comparison in the activity kinetics of these two kinases.

15) For figure 2a/b, it could be interesting to see co-localization and staining of other factors such as 53BP1 or other markers of DNA repair to verify the localization pattern.

Based on reviewer's mention of original Figure 2a/b (now Figure 2c, d), which showed localization of phosphorylated PP4R3 β throughout mitosis, we presume that the reviewer wishes to know the localization of 53BP1 (and other factors for DNA end break repair) specifically during mitosis. The DNA damage response (DDR) during mitosis and localization of DNA repair proteins is a complex and intriguing problem. It has been known since the 1950's (Zirkle and Bloom, 1953 Science) that there is no DNA damage checkpoint in mitosis, and damaged

DNA remains unrepaired. The implication is that DSB repair machinery is not 'active' during mitosis (reviewed very nicely "The DNA damage response during mitosis" Mutat Res. 2013) and consistent with this notion factors such as 53BP1 are not recruited to DSBs in mitosis (Giunta et al. 2010 JCB, Steve Jackson lab). 53BP1 remains in the nuclear soluble and does not localize to DSBs during mitosis. This was the primary content of our previous work (Lee et al, Molecular cell, 2014) and this concept was reviewed by Stephen Jackson and colleagues ("Keeping 53BP1 out of focus in mitosis." Cell Res. 2014 Jul;24(7):781-2.). Since we already know that 53BP1 has a diffuse nuclear staining pattern during mitosis with no discernible foci formation, we did not think it would be useful to examine its co-localization with PP4R3 β . Also, here we examine the localization of PP4R3 β in the absence of DNA damage during mitosis, therefore we do not have any scientific rationale to look at other DDR factors. For example, γ -H2AX, the most well studied marker for DNA damage, will not be formed without DNA damage. While we agree with the reviewer that the localization of DDR proteins during mitosis is an interesting question to address, in this study, we aimed to answer how 53BP1 function is restored for DDR in G1. The broader question of how DDR proteins are localized in mitosis are beyond the scope of our current study.

16) The text needs to be improved as it contains many typos and sentences that are unclear.

We have edited the manuscript for clarity and attempted to correct all the typos.

REVIEWERS' COMMENTS:

Reviewer #1 (Remarks to the Author):

I appreciate that the authors addressed all of my major and minor concerns. My major concern, the timing between CDK5 activity, S840 phosphorylation of PP4R3 β , and T1609 and S1618 phosphorylation of 53BP1 from G2/M to G1 phase are now clear and consistent between three different cell lines, resulted in clearer conclusion model in Figure 5e. I think the manuscript was significantly improved by the revision, and the proposed novel mechanism about how DNA damage response is attenuated in mitosis will attract attention in the field of mitosis and DNA damage.

One suggestion is that it is better to include "mitosis" somewhere in the title. The entire studies are specific regulation of the DDR pathway during mitosis, and current title will mislead the readers into thinking that authors find new DDR pathway in general.

Reviewer #2 (Remarks to the Author):

The authors have addressed my concerns and I recommend this manuscript for publication.

REVIEWERS' COMMENTS:

Reviewer #1 (Remarks to the Author):

I appreciate that the authors addressed all of my major and minor concerns. My major concern, the timing between CDK5 activity, S840 phosphorylation of PP4R3 β , and T1609 and S1618 phosphorylation of 53BP1 from G2/M to G1 phase are now clear and consistent between three different cell lines, resulted in clearer conclusion model in Figure 5e. I think the manuscript was significantly improved by the revision, and the proposed novel mechanism about how DNA damage response is attenuated in mitosis will attract attention in the field of mitosis and DNA damage.

One suggestion is that it is better to include “mitosis” somewhere in the title. The entire studies are specific regulation of the DDR pathway during mitosis, and current title will mislead the readers into thinking that authors find new DDR pathway in general.

We thank Reviewer 1 for the suggestion and have modified the title of the manuscript to “A mitotic CDK5-PP4 phospho-signaling cascade primes 53BP1 for DNA repair in G1.”

Reviewer #2 (Remarks to the Author):

The authors have addressed my concerns and I recommend this manuscript for publication.

We thank the reviewers again for their invaluable comments and suggestions. We are delighted to be able to address reviewers' concerns to their satisfaction such that they can enthusiastically recommend our work for publication.